# Toolformer:
# Language Models Can Teach Themselves to Use Tools

**Timo Schick**   **Jane Dwivedi-Yu**   **Roberto Dessì**[†]   **Roberta Raileanu**

**Maria Lomeli**   **Eric Hambro**   **Luke Zettlemoyer**   **Nicola Cancedda**   **Thomas Scialom**

FAIR, Meta  [†]Universitat Pompeu Fabra

## Abstract

Language models (LMs) exhibit remarkable abilities to solve new tasks from just a few examples or textual instructions, especially at scale. They also, paradoxically, struggle with basic functionality, such as arithmetic or factual lookup, where much simpler and smaller specialized models excel. In this paper, we show that LMs can teach themselves to *use external tools* via simple APIs and achieve the best of both worlds. We introduce *Toolformer*, a model trained to decide which APIs to call, when to call them, what arguments to pass, and how to best incorporate the results into future token prediction. This is done in a self-supervised way, requiring nothing more than a handful of demonstrations for each API. We incorporate a range of tools, including a calculator, a Q&A system, a search engine, a translation system, and a calendar. Toolformer achieves substantially improved zero-shot performance across a variety of downstream tasks, often competitive with much larger models, without sacrificing its core language modeling abilities.

## 1  Introduction

Large language models achieve impressive zero and few-shot results on a variety of natural language processing tasks (Brown et al., 2020; Chowdhery et al., 2022, i.a.). However, these models have several inherent limitations that can at best be partially addressed by further scaling. These limitations include an inability to access up-to-date information on recent events (Komeili et al., 2022) and the related tendency to hallucinate facts (Maynez et al., 2020; Ji et al., 2022), difficulties in understanding low-resource languages (Lin et al., 2021), a lack of mathematical skills to perform precise calculations (Patel et al., 2021) and an unawareness of the progression of time (Dhingra et al., 2022).

A simple way to overcome the limitations of today's language models is to give them the ability to *use external tools* such as search engines, calculators, or calendars. However, existing approaches either rely on large amounts of human annotations (Komeili et al., 2022; Thoppilan et al., 2022) or limit tool use to task-specific settings only (e.g., Gao et al., 2022; Parisi et al., 2022), hindering a more widespread adoption of tool use in LMs. Therefore, we propose *Toolformer*, a model that learns to use tools in a novel way, which fulfills the following desiderata:

- Tool use should be learned in a self-supervised way without large amounts of *human annotations*. This is important not only because of the costs associated with such annotations, but also because what humans find useful may be different from what a model finds useful.

- The LM should not lose any of its *generality* and should be able to decide for itself *when* and *how* to use which tool. In contrast to existing approaches, this enables a much more comprehensive use of tools that is not tied to specific tasks.

Our approach for achieving these goals is based on the recent idea of using large LMs with *in-context learning* (Brown et al., 2020) to generate entire datasets from scratch (Schick and Schütze, 2021b;

37th Conference on Neural Information Processing Systems (NeurIPS 2023).

The New England Journal of Medicine is a registered trademark of **[QA("Who is the publisher of The New England Journal of Medicine?") → Massachusetts Medical Society]** the MMS.

Out of 1400 participants, 400 (or **[Calculator(400 / 1400) → 0.29]** 29%) passed the test.

The name derives from "la tortuga", the Spanish word for **[MT("tortuga") → turtle]** turtle.

The Brown Act is California's law **[WikiSearch("Brown Act") → The Ralph M. Brown Act is an act of the California State Legislature that guarantees the public's right to attend and participate in meetings of local legislative bodies.]** that requires legislative bodies, like city councils, to hold their meetings open to the public.

Figure 1: Exemplary predictions of Toolformer. The model autonomously decides to call different APIs (from top to bottom: a question answering system, a calculator, a machine translation system, and a Wikipedia search engine) to obtain information that is useful for completing a piece of text.

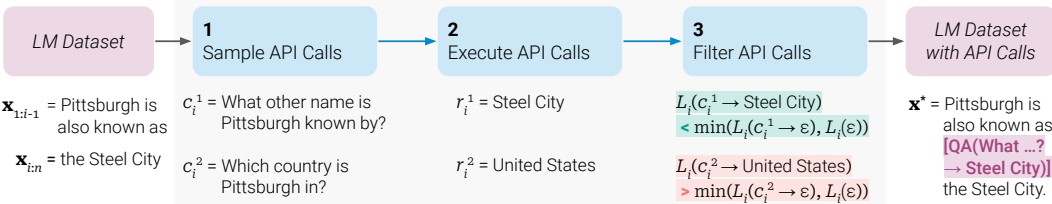

Figure 2: Key steps in our approach, illustrated for a *question answering* tool: Given an input text $\mathbf{x}$, we first sample a position $i$ and corresponding API call candidates $c_i^1, c_i^2, \ldots, c_i^k$. We then execute these API calls and filter out all calls which do not reduce the loss $L_i$ over the next tokens. All remaining API calls are interleaved with the original text, resulting in a new text $\mathbf{x}^*$.

Honovich et al., 2022; Wang et al., 2022): Given just a handful of human-written examples of how an API can be used, we let a LM annotate a huge language modeling dataset with potential API calls. We then use a self-supervised loss to determine which of these API calls actually help the model in predicting future tokens. Finally, we finetune the LM itself on the API calls that it considers useful. As illustrated in Figure 1, through this simple approach, LMs can learn to control a variety of tools, and to choose for themselves which tool to use when and how.

As our approach is agnostic of the dataset being used, we can apply it to the exact same dataset that was used to pretrain a model in the first place. This ensures that the model does not lose any of its generality and language modeling abilities. We conduct experiments on a variety of different downstream tasks, demonstrating that after learning to use tools, Toolformer, which is based on a pretrained GPT-J model (Wang and Komatsuzaki, 2021) with 6.7B parameters, achieves much stronger zero-shot results, clearly outperforming a much larger GPT-3 model (Brown et al., 2020) and several other baselines on various tasks.

## 2 Approach

Our aim is to equip a language model $M$ with the ability to use different tools through API calls. We represent API calls as tuples $c = (a_c, i_c)$ where $a_c$ is the name of the API and $i_c$ is the corresponding input. Given an API call $c$ with a corresponding result $r$, we denote the linearized sequences of the API call not including and including its result, respectively, as:

$$\mathrm{e}(c) = \texttt{<API>}\, a_c(i_c)\, \texttt{</API>} \qquad \mathrm{e}(c, r) = \texttt{<API>}\, a_c(i_c) \to r\, \texttt{</API>}$$

where "`<API>`", "`</API>`" and "→" are special tokens.[1] Some examples of linearized API calls inserted into text sequences are shown in Figure 1.

---

[1] In practice, we use the token sequences " [", "]" and "->" to represent "`<API>`", "`</API>`" and "→", respectively. This enables our approach to work without modifying the existing LM's vocabulary. For reasons of readability, we still refer to them as "`<API>`", "`</API>`" and "→" throughout this section.

*Your task is to add calls to a Question Answering API to a piece of text. The questions should help you get information required to complete the text. You can call the API by writing "[QA(question)]" where "question" is the question you want to ask. Here are some examples of API calls:*

**Input:** Joe Biden was born in Scranton, Pennsylvania.
**Output:** Joe Biden was born in **[QA("Where was Joe Biden born?")]** Scranton, **[QA("In which state is Scranton?")]** Pennsylvania.

**Input:** Coca-Cola, or Coke, is a carbonated soft drink manufactured by the Coca-Cola Company.
**Output:** Coca-Cola, or **[QA("What other name is Coca-Cola known by?")]** Coke, is a carbonated soft drink manufactured by **[QA("Who manufactures Coca-Cola?")]** the Coca-Cola Company.

**Input: x**
**Output:**

Figure 3: An exemplary prompt $P(\mathbf{x})$ used to generate API calls for the question answering tool.

Given a dataset $\mathcal{C} = \{\mathbf{x}^1, \ldots, \mathbf{x}^{|\mathcal{C}|}\}$ of plain texts, we first convert this dataset into a dataset $\mathcal{C}^*$ augmented with API calls. This is done in three steps, illustrated in Figure 2: First, we exploit the in-context learning ability of $M$ to sample a large number of API calls. We then execute them and finally check whether the obtained responses are helpful for predicting future tokens; this is used as a filtering criterion. After filtering, we merge API calls for different tools, resulting in the augmented dataset $\mathcal{C}^*$, and finetune $M$ itself on this dataset. Each step is described in more detail below.

**Sampling API Calls** For each API, we write a prompt $P(\mathbf{x})$ that encourages the LM to annotate an example $\mathbf{x} = x_1, \ldots, x_n$ with API calls. An example of such a prompt for a question answering tool is shown in Figure 3. Let $p_M(z_{n+1} \mid z_1, \ldots, z_n)$ be the probability that $M$ assigns to token $z_{n+1}$ as a continuation for the sequence $z_1, \ldots, z_n$. We first sample up to $k$ candidate *positions* for doing API calls by computing, for each $i \in \{1, \ldots, n\}$, the probability $p_i = p_M(\texttt{<API>} \mid P(\mathbf{x}), x_{1:i-1})$ that $M$ assigns to starting an API call at position $i$. Given a sampling threshold $\tau_s$, we keep all positions $I = \{i \mid p_i > \tau_s\}$; if there are more than $k$ such positions, we only keep the top $k$. For each position $i \in I$, we then obtain up to $m$ API calls $c_i^1, \ldots, c_i^m$ by sampling from $M$ given the sequence $[P(\mathbf{x}), x_1, \ldots, x_{i-1}, \texttt{<API>}]$ as a prefix and $\texttt{</API>}$ as an end-of-sequence token.

**Executing API Calls** As a next step, we execute all API calls generated by $M$. How this is done depends entirely on the API itself – for example, it can involve calling another neural network, executing a Python script or using a retrieval system to perform search over a large corpus. The response for each API call $c_i$ needs to be a single text sequence $r_i$.

**Filtering API Calls** Let $i$ be the position of the API call $c_i$ in the sequence $\mathbf{x} = x_1, \ldots, x_n$, and let $r_i$ be the response from the API. Further, given a sequence $(w_i \mid i \in \mathbb{N})$ of *weights*, let

$$L_i(\mathbf{z}) = -\sum_{j=i}^{n} w_{j-i} \cdot \log p_M(x_j \mid \mathbf{z}, x_{1:j-1})$$

be the weighted cross entropy loss for $M$ over the tokens $x_i, \ldots, x_n$ if the model is prefixed with some text sequence $\mathbf{z}$. We compare two different instantiations of this loss:

$$L_i^+ = L_i(\mathrm{e}(c_i, r_i)) \qquad\qquad L_i^- = \min\left(L_i(\varepsilon), L_i(\mathrm{e}(c_i, \varepsilon))\right)$$

where $\varepsilon$ denotes an empty sequence. The former is the weighted loss over all tokens $x_i, \ldots, x_n$ if the API call and its result are given to $M$ as a prefix;[2] the latter is the minimum of the losses obtained from (i) doing no API call at all and (ii) doing an API call, but not providing the response. Intuitively, an API call is helpful to $M$ if providing it with both the input *and* the output of this call makes it easier for the model to predict future tokens, compared to not receiving the API call at all, or receiving only its input. Given a filtering threshold $\tau_f$, we thus only keep API calls for which $L_i^- - L_i^+ \geq \tau_f$ holds, i.e., adding the API call and its result *reduces* the loss by at least $\tau_f$, compared to not doing any API call or obtaining no result from it.

---

[2] We provide $\mathrm{e}(c_i, r_i)$ as a prefix instead of inserting it at position $i$ because $M$ is not yet finetuned on any examples containing API calls, so inserting it in the middle of $\mathbf{x}$ would interrupt the flow and not align with patterns in the pretraining corpus, thus hurting perplexity.

Table 1: Examples of inputs and outputs for all APIs used.

| API Name | Example Input | Example Output |
|---|---|---|
| Question Answering | Where was the Knights of Columbus founded? | New Haven, Connecticut |
| Wikipedia Search | Fishing Reel Types | Spin fishing > Spin fishing is distinguished between fly fishing and bait cast fishing by the type of rod and reel used. There are two types of reels used when spin fishing, the open faced reel and the closed faced reel. |
| Calculator | 27 + 4 * 2 | 35 |
| Calendar | $\varepsilon$ | Today is Monday, January 30, 2023. |
| Machine Translation | sûreté nucléaire | nuclear safety |

**Model Finetuning** After sampling and filtering calls for all APIs, we finally merge the remaining API calls and interleave them with the original inputs. That is, for an input text $\mathbf{x} = x_1, \ldots, x_n$ with a corresponding API call and result $(c_i, r_i)$ at position $i$, we construct the new sequence $\mathbf{x}^* = x_{1:i-1}, \mathrm{e}(c_i, r_i), x_{i:n}$; we proceed analogously for texts with multiple API calls. Doing this for all $\mathbf{x} \in \mathcal{C}$ results in the new dataset $\mathcal{C}^*$ augmented with API calls. We use $\mathcal{C}^*$ to finetune $M$, using a standard language modeling objective. Crucially, apart from inserted API calls, $\mathcal{C}^*$ contains the exact same texts as $\mathcal{C}$, the original dataset. As a consequence, finetuning $M$ on $\mathcal{C}^*$ exposes it to the same content as finetuning on $\mathcal{C}$. Moreover, as API calls are inserted in exactly those positions and with exactly those inputs that help $M$ predict future tokens, finetuning on $\mathcal{C}^*$ enables the language model to decide when and how to use which tool, based purely on its own feedback.

**Inference** When generating text with $M$ after finetuning with our approach, we perform regular decoding until $M$ produces the "→" token, indicating that it next expects the response for an API call. At this point, we interrupt the decoding process, call the appropriate API to get a response, and continue the decoding process after inserting both the response and the `</API>` token.

# 3 Tools

We explore various tools to address different shortcomings of LMs. The only constraints we impose are that (i) their inputs and outputs can be represented as texts, and (ii) we can obtain a few demonstrations of their intended use. Concretely, we explore a question answering system, a Wikipedia search engine, a calculator, a calendar, and a machine translation system. Examples for the APIs associated with each of these tools are shown in Table 1. We briefly discuss all tools below.

**Question Answering** Our first tool is a question answering system based on another LM that can answer simple factoid questions. Specifically, we use *Atlas* (Izacard et al., 2022), a retrieval-augmented LM finetuned on Natural Questions (Kwiatkowski et al., 2019).

**Calculator** As a second tool, we use a calculator that can perform simple numeric calculations; we only support the four basic arithmetic operations. Results are always rounded to two decimal places.

**Wikipedia Search** Our third tool is a search engine that, given a search term, returns short text snippets from Wikipedia. Compared to our question answering tool, this search enables a model to get more comprehensive information on a subject, but requires it to extract the relevant parts by itself. As our search engine, we use a BM25 retriever (Robertson et al., 1995; Baeza-Yates et al., 1999) that indexes the Wikipedia dump from KILT (Petroni et al., 2021).

**Machine Translation System** Our fourth tool is a machine translation system based on a LM that can translate a phrase from any language into English. More concretely, we use the 600M parameter NLLB (Costa-jussà et al., 2022) as our multilingual machine translation model that works for 200 languages (including low-resource ones). The source language is automatically detected using the *fastText* classifier (Joulin et al., 2016), while the target language is always set to English.

**Calendar** Our final tool is a calendar API that, when queried, returns the current date without taking any input. This provides temporal context for predictions that require some awareness of time.

Table 2: Number of examples with API calls in $\mathcal{C}^*$ for different values of our filtering threshold $\tau_f$.

| API | Number of Examples | | |
| --- | --- | --- | --- |
| | $\tau_f = 0.5$ | $\tau_f = 1.0$ | $\tau_f = 2.0$ |
| Question Answering | 51,987 | 18,526 | 5,135 |
| Wikipedia Search | 207,241 | 60,974 | 13,944 |
| Calculator | 3,680 | 994 | 138 |
| Calendar | 61,811 | 20,587 | 3,007 |
| Machine Translation | 3,156 | 1,034 | 229 |

## 4 Experiments

We investigate whether our approach enables a LM to use tools without any further supervision and to decide for itself when and how to call which tool. To test this, we select a variety of downstream tasks where we assume at least one of the considered tools to be useful, and evaluate performance in zero-shot settings (Section 4.2). Beyond that, we also ensure that our approach does not hurt the model's core LM abilities; we verify this by looking at perplexity on two language modeling datasets (Section 4.3). Finally, we investigate how tool use is affected by model size (Section 4.4).

### 4.1 Experimental Setup

We use a subset of CCNet (Wenzek et al., 2020) as our dataset $\mathcal{C}$ and GPT-J (Wang and Komatsuzaki, 2021) as our language model $M$. To reduce the computational cost of annotating $\mathcal{C}$ with API calls, we define heuristics for some APIs to get a subset of $\mathcal{C}$ for which API calls are more likely to be helpful than for an average text. For example, we only consider texts for the calculator tool if they contain at least three numbers. Details of the heuristics used are given in Appendix A. For obtaining $\mathcal{C}^*$ from $\mathcal{C}$, we perform all steps described in Section 2 and additionally filter out all examples for which all API calls were eliminated in the filtering step.[3] For the weighting function, we use

$$w_t = \frac{\tilde{w}_t}{\sum_{s \in \mathbb{N}} \tilde{w}_s} \text{ with } \tilde{w}_t = \max(0, 1 - 0.2 \cdot t)$$

to make sure that API calls happen close to where the information provided by the API is actually helpful for the model. The thresholds $\tau_s$ and $\tau_f$ are chosen individually for each tool to ensure a sufficient number of examples; see Appendix A for details. Table 2 shows relevant statistics of our final dataset augmented with API calls. We finetune $M$ on $\mathcal{C}^*$ using a batch size of 128 and a learning rate of $1 \cdot 10^{-5}$ with linear warmup for the first 10% of training. Finetuning details are given in Appendix B. In our experiments, we mainly compare GPT-J and the following models:

- **GPT-J + CC**: GPT-J finetuned on $\mathcal{C}$, our subset of CCNet *without* any API calls.
- **Toolformer**: GPT-J finetuned on $\mathcal{C}^*$, our subset of CCNet augmented with API calls.
- **Toolformer (disabled)**: The same model as Toolformer, but API calls are disabled during decoding. This is achieved by manually setting the probability of the `<API>` token to 0.

We additionally compare to OPT (66B) (Zhang et al., 2022) and the original `davinci` variant of GPT-3 (175B) (Brown et al., 2020), two models that are about 10 and 25 times larger than GPT-J.

### 4.2 Downstream Tasks

We evaluate on various downstream tasks considering a prompted zero-shot setup: Models are instructed to solve each task in natural language (see Appendix C), but we provide no examples. This is in contrast to prior work on tool use (e.g., Gao et al., 2022; Parisi et al., 2022), where models are provided with dataset-specific examples of how a tool can be used to solve a concrete task. We choose this more challenging setup as we are interested in seeing whether Toolformer works in precisely those cases where a user does not specify in advance which tools should be used in which way.

---

[3]While this filtering alters the distribution of training examples, we assume that the remaining examples are close enough to the original distribution so that $M$'s language modeling abilities remain unaffected. This assumption is empirically validated in Section 4.3.

Table 3: Results on subsets of LAMA and various benchmarks requiring mathematical reasoning. For LAMA, Toolformer uses the question answering tool for most examples, clearly outperforming all baselines of the same size and achieving results competitive with GPT-3. For the math benchmarks, Toolformer makes extensive use of the calculator tool, clearly outperforming OPT and GPT-3. Best results with a GPT-J based model are shown in bold, best results overall are underlined.

| Model | LAMA | | | Math Benchmarks | | |
|---|---|---|---|---|---|---|
| | SQuAD | Google-RE | T-REx | ASDiv | SVAMP | MAWPS |
| GPT-J | 17.8 | 4.9 | 31.9 | 7.5 | 5.2 | 9.9 |
| GPT-J + CC | 19.2 | 5.6 | 33.2 | 9.6 | 5.0 | 9.3 |
| Toolformer (disabled) | 22.1 | 6.3 | 34.9 | 14.8 | 6.3 | 15.0 |
| Toolformer | **33.8** | **11.5** | **53.5** | **40.4** | **29.4** | **44.0** |
| OPT (66B) | 21.6 | 2.9 | 30.1 | 6.0 | 4.9 | 7.9 |
| GPT-3 (175B) | 26.8 | 7.0 | 39.8 | 14.0 | 10.0 | 19.8 |

We use greedy decoding, but with one modification for Toolformer: We let the model start an API call whenever `<API>` is one of the $k$ most likely tokens. For $k = 1$, this corresponds to regular greedy decoding; we instead use $k = 10$ to increase the disposition of our model to make use of APIs. At the same time, we allow at most one API call per input to make sure the model does not get stuck in a loop where it constantly calls APIs. The effect of these modifications is explored in Appendix E.

**LAMA** We evaluate our models on the SQuAD, Google-RE and T-REx subsets of the LAMA benchmark (Petroni et al., 2019). For each of these subsets, the task is to complete a short statement with a missing fact (e.g., a date or a place). As LAMA was originally designed to evaluate *masked* LMs (e.g., Devlin et al., 2019), we filter out examples where the mask token is not the final token, so that all examples can be processed in a left-to-right fashion. To account for different tokenizations and added complexity from not informing the model that a single word is required, for all models we use a slightly more lenient evaluation criterion than exact match and simply check whether the correct word is within the first five words predicted by the model. As LAMA is based on statements obtained directly from Wikipedia, we prevent Toolformer from using the Wikipedia Search API to avoid giving it an unfair advantage. As shown in Table 3 (left), all GPT-J models without tool use achieve similar performance. Crucially, Toolformer clearly outperforms these baseline models, improving upon the best baseline by 11.7, 5.2 and 18.6 points, respectively. It also clearly outperforms OPT (66B) and GPT-3 (175B), despite both models being much larger. This is achieved because the model independently decides to ask the question answering tool for the required information in almost all cases (98.1%); for only very few examples, it uses a different tool (0.7%) or no tool at all (1.2%).

**Math Benchmarks** We test mathematical abilities on ASDiv (Miao et al., 2020), SVAMP (Patel et al., 2021) and the MAWPS benchmark (Koncel-Kedziorski et al., 2016). We again account for the fact that we test all models in a zero-shot setup by using a more lenient evaluation criterion: As the required output is always a number, we simply check for the first number predicted by the model.[4] Results are shown in Table 3 (right). While GPT-J and GPT-J + CC perform about the same, Toolformer achieves stronger results even without API calls. We surmise that this is because the model is finetuned on many examples of API calls and their results, improving its own mathematical capabilities. Nonetheless, allowing the model to make API calls more than doubles performance for all tasks, and also clearly outperforms the much larger OPT and GPT-3. This is because across all benchmarks, for 97.9% of all examples the model decides to ask the calculator tool for help.

**Question Answering** We look at Web Questions (Berant et al., 2013), Natural Questions (Kwiatkowski et al., 2019) and TriviaQA (Joshi et al., 2017). For evaluation, we check whether the first 20 words predicted by a model contain the correct answer instead of requiring an exact match. For Toolformer, we disable the question answering tool as this would make solving the tasks trivial. Results are shown in Table 4 (left). Once again, Toolformer clearly outperforms all other models based on GPT-J, relying on the Wikipedia search API (99.3%) to find relevant information. However,

---

[4]An exception to this is if the model's prediction contains an equation (e.g., "The correct answer is 5+3=8"), in which case we consider the first number after the "=" sign to be its prediction.

Table 4: Results for various question answering datasets and temporal datasets. Using the Wikipedia search tool for most examples, Toolformer clearly outperforms baselines of the same size, but falls short of GPT-3 (175B) for question answering tasks. For temporal datasets, Toolformer outperforms all baselines, but does not make use of the calendar tool for TEMPLAMA.

| Model | LAMA | | | Temporal Datasets | |
| | WebQS | NQ | TriviaQA | TEMPLAMA | DATESET |
|---|---|---|---|---|---|
| GPT-J | 18.5 | 12.8 | 43.9 | 13.7 | 3.9 |
| GPT-J + CC | 18.4 | 12.2 | 45.6 | 12.9 | 2.9 |
| Toolformer (disabled) | 18.9 | 12.6 | 46.7 | 12.7 | 5.9 |
| Toolformer | **26.3** | **17.7** | **48.8** | **16.3** | **27.3** |
| OPT (66B) | 18.6 | 11.4 | 45.7 | 14.5 | 1.3 |
| GPT-3 (175B) | 29.0 | 22.6 | 65.9 | 15.5 | 0.8 |

Table 5: Results on MLQA for Spanish (Es), German (De), Hindi (Hi), Vietnamese (Vi), Chinese (Zh) and Arabic (Ar). While using the MT tool to translate questions is helpful across all languages, further pretraining on CCNet deteriorates performance; thus, Toolformer does not consistently outperform GPT-J. The final rows correspond to models that are given contexts and questions in English.

| Model | Es | De | Hi | Vi | Zh | Ar |
|---|---|---|---|---|---|---|
| GPT-J | 15.2 | **16.5** | 1.3 | 8.2 | **18.2** | **8.2** |
| GPT-J + CC | 15.7 | 14.9 | 0.5 | 8.3 | 13.7 | 4.6 |
| Toolformer (disabled) | 19.8 | 11.9 | 1.2 | 10.1 | 15.0 | 3.1 |
| Toolformer | **20.6** | 13.5 | **1.4** | **10.6** | 16.8 | 3.7 |
| OPT (66B) | 0.3 | 0.1 | 1.1 | 0.2 | 0.7 | 0.1 |
| GPT-3 (175B) | 3.4 | 1.1 | 0.1 | 1.7 | 17.7 | 0.1 |
| GPT-J (All En) | 24.3 | 27.0 | 23.9 | 23.3 | 23.1 | 23.6 |
| GPT-3 (All En) | 24.7 | 27.2 | 26.1 | 24.9 | 23.6 | 24.0 |

Toolformer still lags behind the much larger GPT-3 (175B) model. This is likely due to both the simplicity of our search engine (in many cases, it returns results that are clearly not a good match for a given query) and the inability of Toolformer to *interact* with it, e.g., by reformulating its query if results are not helpful or by browsing through multiple of the top results.

**Multilingual QA**  We evaluate all models on MLQA (Lewis et al., 2019), a multilingual QA benchmark. Context for each question is provided in English, while the question can be in Arabic, German, Spanish, Hindi, Vietnamese, or Simplified Chinese. Our evaluation metric is the percentage of times the model's generation, capped at 10 words, contains the correct answer. Results are shown in Table 5. API calls improve Toolformer's performance for all languages, suggesting that it has learned to make use of the machine translation tool. Depending on the language, this tool is used for 63.8% to 94.9% of all examples; the only exception is Hindi, for which it is used in only 7.3% of cases. However, Toolformer does not consistently outperform GPT-J as finetuning on CCNet deteriorates performance for some languages. OPT and GPT-3 perform surprisingly weak across all languages, mostly because they fail to provide an answer in English despite being instructed to do so. A potential reason for GPT-J not suffering from this problem is that it was trained on more multilingual data than both OPT and GPT-3, including EuroParl (Koehn, 2005). As an upper bound, we also evaluate GPT-J and GPT-3 on a variant of MLQA where both the context and the question are provided in English. In this setup, GPT-3 performs better than all other models, supporting our hypothesis that its subpar performance on MLQA is due to the task's multilingual aspect.

**Temporal Datasets**  We evaluate all models on TEMPLAMA (Dhingra et al., 2022) and a new dataset that we call DATESET. TEMPLAMA contains cloze queries about facts that change with time. DATESET, described in Appendix D, is generated through a series of templates, but populated using a combination of random dates/durations (e.g., "What day of the week was it 30 days ago?"). For both tasks, we use the same evaluation as for the original LAMA dataset. Results shown in Table 4 (right) illustrate that Toolformer outperforms all baselines for both TEMPLAMA and DATESET. However,

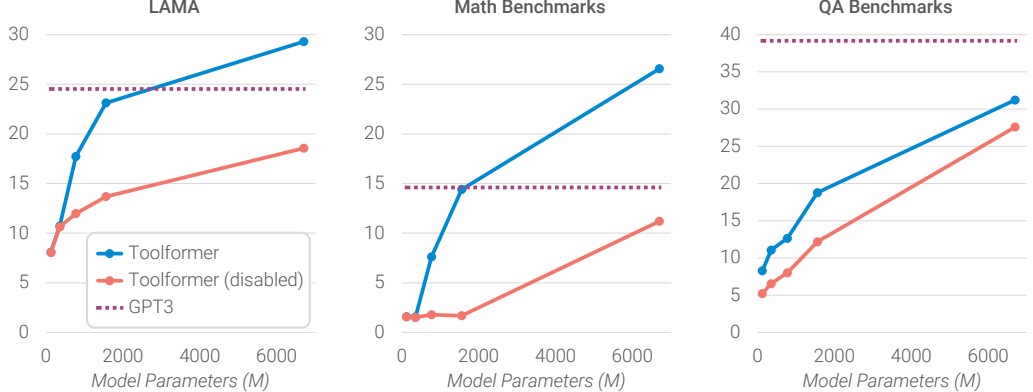

Figure 4: Average performance on LAMA, our math benchmarks and our QA benchmarks for GPT-2 models of different sizes and GPT-J finetuned with our approach, both with and without API calls. While API calls are not helpful to the smallest models, larger models learn how to make good use of them. Even for bigger models, the gap between predictions with and without API calls remains high.

closer inspection shows that improvements on TEMPLAMA can not be attributed to the calendar tool, which is only used for 0.2% of all examples, but mostly to the Wikipedia search and question answering tools.This makes sense given that entities in TEMPLAMA are often so specific and rare that even knowing the date alone would be of little help. The best course of action for this dataset – first querying the calendar API to get the current date, and then querying the QA system with this date – is not only prohibited by our restriction of using at most one API call, but also hard to learn for Toolformer given that all API calls in its training data are sampled independently. For DATESET, on the other hand, the considerable improvement of Toolformer compared to other models can be fully accredited to the calendar tool, which it makes use of for 54.8% of all examples.

### 4.3 Language Modeling

We want to ensure that language modeling performance of Toolformer does not degrade through finetuning with API calls. To this end, we evaluate our models on two language modeling datasets: WikiText (Merity et al., 2017) and a subset of 10,000 randomly selected documents from CCNet (Wenzek et al., 2020) that were not used during training. Finetuning on CCNet leads to slightly improved performance on the CCNet evaluation subset (perplexity improves from 10.6 to 10.5), but slightly deteriorates performance on WikiText (9.9 to 10.3), presumably because the original pretraining data for GPT-J is more similar to WikiText than our subset of CCNet. Most importantly, however, training on $\mathcal{C}^*$ does not lead to an increase in perplexity compared to training on $\mathcal{C}$ when API calls are disabled at inference time, giving perplexities of 10.5 and 10.3, respectively.[5]

### 4.4 Scaling Laws

We investigate how the ability to ask external tools for help affects performance as we vary the size of our LM. To this end, we apply our approach not just to GPT-J, but also to four smaller models from the GPT-2 family (Radford et al., 2019), with 124M, 355M, 775M and 1.6B parameters, respectively. We do so using only a subset of three tools: the question answering system, the calculator, and the Wikipedia search engine. Apart from this, we follow the experimental setup described in Section 4.1. Figure 4 shows that the ability to leverage the provided tools only emerges at around 775M parameters: smaller models achieve similar performance both with and without tools. An exception to this is the Wikipedia search engine used mostly for QA benchmarks; we hypothesize that this is because the API is comparably easy to use. While models become better at solving tasks *without* API calls as they grow in size, their ability to make good use of the provided API improves at the same time. Thus, there remains a large gap between predictions with and without API calls even for our biggest model.

---

[5]We do not evaluate the perplexity of Toolformer with API calls enabled as computing the probability $p_M(x_t \mid x_1, \ldots, x_{t-1})$ of token $x_t$ given $x_1, \ldots, x_{t-1}$ would require marginalizing over all potential API calls that the model could make at position $t$, which is intractable.

In Appendix G, we extend these investigations in scale to the LLaMA v1 7B model (Touvron et al., 2023) to see how tool-use scales with model capability instead of size. We find that the utility of weaker tools such as the WikiSearch tool vanish for these stronger base-models, and we demonstrate the value of generating and scoring with a strong model, compared to simply finetuning.

## 5   Related Work

**Language Model Pretraining**   There are various approaches that augment LMs with some form of additional textual information during pretraining, including various forms of metadata (Keskar et al., 2019), HTML tags (Aghajanyan et al., 2021), Wikipedia markup (Schick et al., 2022), or related texts obtained from an information retrieval system (Guu et al., 2020; Borgeaud et al., 2021; Izacard et al., 2022). For all of these approaches, additional information is *always* provided, regardless of whether it is helpful or not. In contrast, Toolformer learns for itself to explicitly asks for the right information.

**Tool Use**   Several approaches aim to equip LMs with the ability to use external tools such as search engines (Komeili et al., 2022; Thoppilan et al., 2022; Lazaridou et al., 2022; Shuster et al., 2022; Yao et al., 2022), web browsers (Nakano et al., 2021), calculators (Cobbe et al., 2021; Thoppilan et al., 2022), translation systems (Thoppilan et al., 2022) and Python interpreters (Gao et al., 2022). The way these models learn to use tools can roughly be divided into two approaches: Either they rely on large amounts of human supervision (Komeili et al., 2022; Nakano et al., 2021; Thoppilan et al., 2022) or they work by prompting the language model in a few-shot setup tailored towards a specific task where it is known a priori which tools needs to be used (Gao et al., 2022; Lazaridou et al., 2022; Yao et al., 2022). In contrast, the self-supervised nature of Toolformer enables it to learn how and when to use tools without requiring a specific prompt that shows task-specific examples of how a tool could be used. Perhaps most closely related to our work is TALM (Parisi et al., 2022), an approach that uses a similar self-supervised objective for teaching a model to use a calculator and a search engine, but explores this only in settings where a model is finetuned for downstream tasks.

**Bootstrapping**   The idea of using self-training and bootstrapping techniques to improve models has been investigated in various contexts, ranging from word sense disambiguation (Yarowsky, 1995), relation extraction (Brin, 1999; Agichtein and Gravano, 2000), parsing (McClosky et al., 2006), sequence generation (He et al., 2020), few-shot text classification (Schick and Schütze, 2021a) and retrieval (Izacard and Grave, 2021) to reasoning (Zelikman et al., 2022). In a similar spirit, Toolformer is trained on its own predictions after applying a perplexity-based filtering step.

## 6   Limitations

While our approach enables LMs to learn how to use a variety of tools in a self-supervised way, there are some clear limitations to what can be achieved with our method in its current form. One such limitation is the inability of Toolformer to use tools in a *chain* (i.e., using the output of one tool as an input for another tool). This is due to the fact that API calls for each tool are generated independently; as a consequence, there are no examples of chained tool use in the finetuning dataset, since this would necessitate multiple API calls per example. Our current approach also does not allow the LM to use a tool in an *interactive* way – especially for tools such as search engines, that could potentially return hundreds of different results, enabling a LM to browse through these results or to refine its search query in a similar spirit to Nakano et al. (2021) can be crucial for certain applications. Beyond this, we found models trained with Toolformer to often be sensitive to the exact wording of their input when deciding whether or not to call an API; this is perhaps unsurprising given that LMs are known to be very sensitive to the prompt they are provided with in both zero- and few-shot settings (Jiang et al., 2020; Schick and Schütze, 2021a). Depending on the tool, our method is also very sample-inefficient; for example, processing more than a million documents results in only a few thousand examples of useful calls to the calculator API. A potential solution to this problem might be to iteratively apply our approach, similar to how this is done in related bootstrapping approaches (Schick and Schütze, 2021a; Izacard and Grave, 2021; Parisi et al., 2022). Finally, when deciding whether or not to make an API call, Toolformer currently does not take into account the tool-dependent, computational cost incurred from making an API call.

# 7 Conclusion

We have introduced Toolformer, a LM that learns in a self-supervised way how to use different tools such as search engines, calculators, and translation systems via simple API calls. This is done by finetuning on sampled API calls that are filtered based on whether they reduce perplexity on future tokens. Toolformer considerably improves zero-shot performance of a 6.7B parameter GPT-J model, enabling it to even outperform a much larger GPT-3 model on a range of different downstream tasks.

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
