# A   API Details

When sampling and filtering API calls, by default we use values of $\tau_s = 0.05$ and $\tau_f = 1.0$ – i.e., we only make API calls at positions where the probability of the `<API>` token is at least 5%, and we keep API calls if they reduce the loss by at least 1.0. We only keep the top $k = 5$ such positions and sample up to $m = 5$ API calls for each position identified in a piece of text. Due to the heuristic filtering described below, we generate API calls for the calculator and machine translation system on only a small subset of $\mathcal{C}$; to compensate for this, we set $\tau_s = 0.0$, $k = 20$ and $m = 10$ for these tools. As the resulting sets of API calls are still comparably small, we additionally set $\tau_f = 0.5$.

## A.1   Implementation

**Question Answering**   We use the Atlas model of Izacard et al. (2022) finetuned on Natural Questions (Kwiatkowski et al., 2019) as our question answering system. For creating $\mathcal{C}^*$ we use Atlas-large, enabling us to efficiently process millions of API calls; during inference, we use the larger Atlas-xxl model.

**Calculator**   Our calculator is based on a simple Python script and only supports the operators "+", "−", "∗", and "/". It does not return any result for syntactically invalid equations. For sampling API calls, we apply heuristic filters to our subset of CCNet and only process documents that either (i) contain at least three numbers within a window of 100 tokens, where one of these numbers is the result of applying a mathematical operation to the other two, (ii) contain one of the sequences "=", "equals", "equal to", "total of", "average of" followed by a number, or (iii) contain at least three numbers; for texts that only match the last criterion, we only keep a random subset of 1%.

**Calendar**   For creating our dataset $\mathcal{C}^*$, we operate under the assumption that the calendar date in such cases should be the date that the document was created. We approximate this by extracting the date from the URL, if it is present. We filter out texts for which a date cannot be extracted, leaving around 18% of the documents.

**Machine Translation**   For both training and inference, we use the 600M parameter NLLB (Costa-jussà et al., 2022) as our machine translation (MT) model. The source language is automatically detected using the fastText classifier (Joulin et al., 2016), while the target language is always set to English. Since most of the CCNet dataset is in English, we filter out the parts that contain only English text before generating API calls. More specifically, we only keep those paragraphs which contain text chunks in a language other than English preceded and followed by English text. We use text chunks of size 10 tokens. To determine whether the middle text chunk is in a language different than English we again use the fastText classifier with a confidence greater than 0.8. We also filter out any text chunks that contain only numbers or special symbols. This filtering mechanism allows us to generate data more efficiently by focusing our API call generations in places where the MT tool is likely to be helpful. After generating the MT API calls, we additionally remove from our training set those where the input to the MT tool appears after the API call but not before it. While during data generation the model can look ahead to generate API calls, this is not possible at inference time, so we want to dissuade the model from calling the API in such cases.

## A.2   Prompts

Below, we list the prompts used to sample API calls for each tool considered.

**Question Answering**   We use the following prompt for the question answering tool:

```
Your task is to add calls to a Question Answering API to a piece of text. The
questions should help you get information required to complete the text. You can
call the API by writing "[QA(question)]" where "question" is the question you want
to ask. Here are some examples of API calls:
Input: Joe Biden was born in Scranton, Pennsylvania.
Output: Joe Biden was born in [QA("Where was Joe Biden born?")] Scranton, [QA("In
which state is Scranton?")] Pennsylvania.

```

```
50  Input: Coca-Cola, or Coke, is a carbonated soft drink manufactured by the Coca-Cola
51  Company.
52  Output: Coca-Cola, or [QA("What other name is Coca-Cola known by?")] Coke, is
53  a carbonated soft drink manufactured by [QA("Who manufactures Coca-Cola?")] the
54  Coca-Cola Company.
55
56  Input: x
57  Output:
```

**Calculator**   We use the following prompt for the calculator:

```
59  Your task is to add calls to a Calculator API to a piece of text. The calls should
60  help you get information required to complete the text. You can call the API by
61  writing "[Calculator(expression)]" where "expression" is the expression to be
62  computed. Here are some examples of API calls:
63  Input: The number in the next term is 18 + 12 x 3 = 54.
64  Output: The number in the next term is 18 + 12 x 3 = [Calculator(18 + 12 * 3)] 54.
65
66  Input: The population is 658,893 people. This is 11.4% of the national average of
67  5,763,868 people.
68  Output: The population is 658,893 people. This is 11.4% of the national average of
69  [Calculator(658,893 / 11.4%)] 5,763,868 people.
70
71  Input: A total of 252 qualifying matches were played, and 723 goals were scored (an
72  average of 2.87 per match). This is twenty goals more than the 703 goals last year.
73  Output: A total of 252 qualifying matches were played, and 723 goals were scored (an
74  average of [Calculator(723 / 252)] 2.87 per match). This is twenty goals more than
75  the [Calculator(723 - 20)] 703 goals last year.
76
77  Input: I went to Paris in 1994 and stayed there until 2011, so in total, it was 17
78  years.
79  Output: I went to Paris in 1994 and stayed there until 2011, so in total, it was
80  [Calculator(2011 - 1994)] 17 years.
81
82  Input: From this, we have 4 * 30 minutes = 120 minutes.
83  Output: From this, we have 4 * 30 minutes = [Calculator(4 * 30)] 120 minutes.
84
85  Input: x
86  Output:
```

**Wikipedia Search**   We use the following prompt for the Wikipedia search tool:

```
88  Your task is to complete a given piece of text. You can use a Wikipedia Search API
89  to look up information. You can do so by writing "[WikiSearch(term)]" where "term"
90  is the search term you want to look up. Here are some examples of API calls:
91  Input: The colors on the flag of Ghana have the following meanings: red is for the
92  blood of martyrs, green for forests, and gold for mineral wealth.
93  Output: The colors on the flag of Ghana have the following meanings: red is for
94  [WikiSearch("Ghana flag red meaning")] the blood of martyrs, green for forests, and
95  gold for mineral wealth.
96
97  Input: But what are the risks during production of nanomaterials? Some nanomaterials
98  may give rise to various kinds of lung damage.
99  Output: But what are the risks during production of nanomaterials?
100 [WikiSearch("nanomaterial production risks")] Some nanomaterials may give rise
101 to various kinds of lung damage.
102
103 Input: Metformin is the first-line drug for patients with type 2 diabetes and
104 obesity.
105 Output: Metformin is the first-line drug for [WikiSearch("Metformin first-line
106 drug")] patients with type 2 diabetes and obesity.
107
108 Input: x
109 Output:
```

**Machine Translation**  We use the following prompt for the machine translation tool:

```
Your task is to complete a given piece of text by using a Machine Translation API.
You can do so by writing "[MT(text)]" where text is the text to be translated into
English.
Here are some examples:

Input: He has published one book: O homem suprimido ("The Supressed Man")
Output: He has published one book: O homem suprimido [MT(O homem suprimido)] ("The
Supressed Man")

Input: In Morris de Jonge's Jeschuah, der klassische jüdische Mann, there is a
description of a Jewish writer
Output: In Morris de Jonge's Jeschuah, der klassische jüdische Mann [MT(der
klassische jüdische Mann)], there is a description of a Jewish writer

Input: 南京高淳县住房和城乡建设局 城市新区设计 a plane of reference Gaochun is one of
seven districts of the provincial capital Nanjing
Output: [MT(南京高淳县住房和城乡建设局 城市新区设计)] a plane of reference Gaochun is
one of seven districts of the provincial capital Nanjing

Input: x
Output:
```

**Calendar**  We use the following prompt for the calendar tool:

```
Your task is to add calls to a Calendar API to a piece of text. The API calls should
help you get information required to complete the text. You can call the API by
writing "[Calendar()]" Here are some examples of API calls:

Input: Today is the first Friday of the year.
Output: Today is the first [Calendar()] Friday of the year.

Input: The president of the United States is Joe Biden.
Output: The president of the United States is [Calendar()] Joe Biden.

Input: The current day of the week is Wednesday.
Output: The current day of the week is [Calendar()] Wednesday.

Input: The number of days from now until Christmas is 30.
Output: The number of days from now until Christmas is [Calendar()] 30.

Input: The store is never open on the weekend, so today it is closed.
Output: The store is never open on the weekend, so today [Calendar()] it is closed.

Input: x
Output:
```

## B Toolformer Training

We use up to 25k examples per API and train with a maximum sequence length of 1,024, using an effective batch size of 128. All models are trained using DeepSpeed's ZeRO-3 (Rasley et al., 2020) on 8 NVIDIA A100 40GB GPUs with BF16. Training is performed for up to 2,000 steps (requiring approximately 10h), where we evaluate perplexity on a small development set from CCNet containing 1,000 examples every 500 steps and pick the checkpoint that performs best on this development set.

## C Zero-Shot Prompts

### C.1 LAMA and TEMPLAMA

For both LAMA and TEMPLAMA, given an input text x, we use the following prompt: `Please complete the following text so that it is factually correct:  x.`

Table 1: Templates used to create DATESET where a *current_date* is randomly selected. For each *current_date*, a random *past_date* and *future_date* is generated and used to fill each template, if relevant. The federal holidays in the United States (e.g., Thanksgiving) were used in the templates involving holidays.

| Template | Size |
|---|---|
| How many days {ago was, are there until} {*past_date*, *future_date*}? | 400 |
| What {day of the week, day of the month, month, year} was it (*current_date – past_date*) {days, weeks, months, years} ago? | 800 |
| What {day of the week, day of the month, month, year} will it be in (*future_date – current_date*) days? | 800 |
| What day of the week {is, was} it on {*past_date*, *future_date*}? | 400 |
| What {day of the week, day of the month, month, year} {is, was} it {the day before yesterday, yesterday, today, tomorrow, the day after tomorrow}? | 4,000 |
| What {day of the week, day of the month, month} {is, was} $holiday$ this year? | 1,800 |
| How many {days, weeks, months, years} {ago was, are there until} $holiday$ this year? | 1,200 |
| Total | 9,400 |

Table 2: Perplexities of different models on WikiText and our validation subset of CCNet. Adding API calls comes without a cost in terms of perplexity for language modeling without any API calls.

| Model | WikiText | CCNet |
|---|---|---|
| GPT-J | **9.9** | 10.6 |
| GPT-J + CC | 10.3 | **10.5** |
| Toolformer (disabled) | 10.3 | **10.5** |

## C.2 Math Benchmarks

For all math benchmarks, given a context **x** and a question **q**, our prompt is: **x q** `The answer is`.

## C.3 Question Answering

For all question answering datasets, including DATESET, we simply prefix the question with `Answer the following question:`. We append a question mark if the question does not already end with one.

## C.4 Multilingual Question Answering

For MLQA, given a context **x** and a question **q**, our prompt is: `Your task is to answer a question based on the following paragraph:` **x** `Now answer the following question in English:` **q**.

## C.5 LLaMA Question Answering

For question answering in LLaMA + Toolformer investigations in Appendix G, we use the following prompt: `"Q: {question}\nA:"` . This prompt is standardly used in these tasks, and was used for the LLaMA toolformer investigations.

## D DATESET

DATESET is created by first randomly selecting 500 "current dates". For each current date, another relatively past/future date is randomly selected within a four-year range, and the two dates are used to fill the query templates in Table 1. An example of one such query using the first template would be, "How many days ago was August 14, 2020?" If called, the Calendar tool would return the presumed current date (e.g., "Today is Sunday, November 20, 2020").

Table 3: Toolformer results on the T-REx subset of LAMA and on WebQS for different values of $k$ used during decoding. Numbers shown are overall performance (All), performance on the subset where the model decides to make an API call (AC) and all remaining examples (NC), as well as the percentage of examples for which the model decides to call an API (%).

| | T-REx | | | | WebQS | | | |
|---|---|---|---|---|---|---|---|---|
| $k$ | **All** | **AC** | **NC** | **%** | **All** | **AC** | **NC** | **%** |
| 0 | 34.9 | – | 34.9 | 0.0 | 18.9 | – | 18.9 | 0.0 |
| 1 | 47.8 | 53.0 | 44.3 | 40.3 | 19.3 | 17.1 | 19.9 | 8.5 |
| 3 | 52.9 | 58.0 | 29.0 | 82.8 | **26.3** | 26.5 | 6.6 | 99.3 |
| 10 | **53.5** | 54.0 | 22.5 | 98.1 | **26.3** | 26.4 | – | 100.0 |

## E  Additional Analysis

**Decoding Strategy**  We investigate the effect of our modified decoding strategy introduced in Section 4, where instead of always generating the most likely token, we generate the <API> token if it is one of the $k$ most likely tokens. Table 3 shows performance on the T-REx subset of LAMA and on WebQS for different values of $k$. As expected, increasing $k$ leads to the model doing API calls for more examples – from 40.3% and 8.5% with $k = 1$ (i.e., regular greedy decoding) to 98.1% and 100% for $k = 10$. While for T-REx, there is already a clear improvement in performance with greedy decoding, on WebQS our model only starts to make a substantial number of API calls as we slightly increase $k$. Interestingly, for $k = 1$ the model is calibrated to some extent: It decides to call APIs for examples that it would perform particularly badly on without making API calls. This can be seen from the fact that performance on examples where it decides *not* to make an API call (44.3 and 19.9) is higher than average performance if no API calls are made at all (34.9 and 18.9). However, this calibration is lost for higher values of $k$.

**Data Quality**  We qualitatively analyze some API calls generated with our approach for different APIs. Table 4 shows some examples of texts from CCNet augmented with API calls, as well as the corresponding score $L_i^- - L_i^+$ that is used as a filtering criterion, and whether the API calls made by the model are intuitively useful in the given context. As can be seen, high values of $L_i^- - L_i^+$ typically correspond to useful API calls, whereas low values correspond to API calls that do not provide any information that is useful for predicting future tokens. There are some exceptions, e.g., an API call for "Fast train success" in the fourth example that does not give any relevant information but still reduces perplexity. However, some amount of noise in the API calls that are not filtered can actually be useful as it forces the model finetuned on $\mathcal{C}^*$ to not always blindly follow the results of each call it makes.

## F  Failure Modes

During our evaluation, we observe that failure can arise from the following: failing to call the tool, calling the wrong tool, calling the tool incorrectly, receiving a useless or wrong result from the tool, or failing to come to the wrong conclusion using the tool results. While a detailed and quantitative classification of each failure for each task and tool would require careful human annotation, we do observe certain broad trends.

**Failing to call the tool**  At evaluation time, we generally see an under-use of tool usage, possibly due to the difference in distribution between our fine-tuning dataset and evaluation datasets. While the former consists of CCNet paragraphs, the latter consists of very short question-answering style queries, in the case of the QA tasks. We note that tools are far from being called for every question, especially where the answer is strongly in-weights.

**Calling the wrong tool or calling the correct tool incorrectly**  We find Toolformer very often calls an appropriate tool, and it can often judge the context correctly. In the maths section, we do observe occasional calls to Question Answering and WikiSearch, likely because examples using these tools are much more frequent in the fine-tuning dataset than those using the calculator tool.

| Example | $L_i^- - L_i^+$ | Useful |
|---|---|---|
| The Flodden Window (a war memorial dedicated to The Middleton Archers), in the Grade I-listed Church of St Leonard in Middleton is said to be the oldest war memorial in the United Kingdom. `<API>` **WikiSearch(War memorial Flodden) → Battle of Flodden > Commemoration > The stained-glass Flodden Window in Middleton Parish Church [. . . ] was constructed by Sir Richard Assheton in memory of the Battle of Flodden and the archers from Middleton who fought in it.** `</API>` Sir Richard Assheton of Middleton (who built St Leonard) was granted knighthood [. . . ] | 5.49 | ✓ |
| Note: The WL will be open on Friday, `<API>` **Calendar() → Today is Thursday, March 9, 2017.** `</API>` March 10, and Sunday, March 19 for regular hours. | 2.11 | ✓ |
| The Nile has an approximate length of `<API>` **QA(What is the approximate length of the Nile?) → 6,853 km** `</API>` 6,853 kilometers, the White Nile being its main source. | 2.08 | ✓ |
| If Venus had an atmosphere similar to Earth's then you would expect Venus' mean temperature to be 499 K (1.74 x 287) rather than 735 K which is `<API>` **Calculator(735 / 499) → 1.47** `</API>` 1.47 (735 / 499) times hotter than it should be. | 1.59 | ✓ |
| You are here: Home / Featured / Catch this fast train to success! `<API>` **WikiSearch(Fast train success) → Fast Train > It also peaked at #23 on the Canadian CHUM singles chart, on June 26, 1971. The success of this single established Myles Goodwyn as the band's main songwriter, and made it possible for April Wine to record a second album.** `</API>` Don't wait weeks and [. . . ] | 0.92 | ✗ |
| Os Melhores Escolas em Jersey 2020 `<API>` **MT(Os Melhores Escolas em Jersey) → The Best Schools in Jersey** `</API>` On this page you can search for Universities, Colleges and Business schools in Jersey | 0.70 | ✓ |
| Enjoy these pictures from the `<API>` **Calendar() → Today is Friday, April 19, 2013.** `</API>` Easter Egg Hunt. | 0.33 | ✓ |
| 85 patients (23%) were hospitalised alive and admitted to a hospital ward. Of them, `<API>` **Calculator(85 / 23) → 3.70** `</API>` 65% had a cardiac aetiology [. . . ] | −0.02 | ✗ |
| But hey, after the `<API>` **Calendar() → Today is Saturday, June 25, 2011.** `</API>` Disneyland fiasco with the fire drill, I think it's safe to say Chewey won't let anyone die in a fire. | −0.41 | ✗ |
| The last time I was with `<API>` **QA(Who was last time I was with?) → The Last Time** `</API>` him I asked what he likes about me and he said he would tell me one day. | −1.23 | ✗ |

Table 4: Examples of API calls for different tools, sorted by the value of $L_i^- - L_i^+$ that is used as a filtering criterion. High values typically correspond to API calls that are intuitively useful for predicting future tokens.

Many of the tools we use cannot be called 'incorrectly', taking either no arguments or strings as inputs—all of which are valid as API calls. In the case of the Calculator tool, at data-augmentation time, we see many incorrect/invalid generations (even with constrained decoding to arithmetic tokens), but since the final dataset contains useful, correct API calls, the fine-tuned model often generates valid calculations. However, these calls are often very simple (often operations involving two numbers), and have low complexity.

**Receiving the wrong result or reasoning failures**  In some cases we see that a tool response is either useless or incorrect—most often this can be seen with the WikiSearch tool which uses a naive BM25 information retrieval algorithm on Wikipedia. The model's response to this varies, sometimes ignoring the result of the tool while at other times incorporating it. More investigation is needed to understand when or why a tool is ignored. Anecdotally, we observe that the natural continuation requires a highly specific answer, but the tool has not returned it—for instance "Harry Styles was born in [WikiSearch(Harry styles) → Harry Styles is a British singer and actor, who has starred in My Policeman] Worcestershire."

Table 5: Number of final datapoints generated by the LLaMA model, compared to the GPT-J model. Note the LLaMA v1 7B WikiSearch was generated with $k = 2$, instead of $k = 5$, to save compute, since the maximum number of points per tool was predetermined to be 25,000.

| | QA | WikiSearch |
|---|---|---|
| *Threshold $\tau_f$* | 1 | 1 |
| *Generated Examples* | | |
| GPT-J | 18,526 | 60,974 |
| LLaMA v1 7B | 32,180 | 22,891 |

## G  LLaMA Base Model Investigations

### G.1  Motivation

Following the contemporaneous release of the LLaMA v1 models with this paper, we subsequently investigated the application of the Toolformer method to the LLaMA v1 7B base model. The purpose of this investigation was to discover to what degree tool-use would improve as model capability improved, in particular examining 2 separate tools: Question Answering (QA) and WikiSearch.

### G.2  Method

We followed the same generation procedure as described in Section 4.1, generating data for each tool from the same datasets, using the same hyperparameters as in Appendix A, albeit with hyperparameter $k = 2$ for the WikiTool (to save compute on generations). We used the same LLaMA v1 base model to score our datasets, and used the same scoring thresholds to select data as for GPT-J. We split these into a development and train set, as with GPT-J, with final quantities of each training dataset shown in Table 5. We then finetuned the model as in Appendix B, albeit with learning rate $2e^{-6}$.

We evaluated the model with the same protocol described in Section 4.2. We used the same prompts as Appendix C.1 for LAMA, and for Question Answering tasks we used the prompt in Appendix C.5. We used this prompt because it is standard in evaluations of QA Tasks, and was used in LLaMA's own evals. The difference in outcomes between using this prompt, and the original one in Section C.3 shows a stark improvement, and can be seen in Table 7.

### G.3  Results

Table 7 shows the performance of a base LLaMA 7B model, evaluated using the protocol in Section 4.2, and prompts from Appendix C. These show that the LLaMA base model is much more generally capable than GPT-J, especially when using the correct prompts, outperforming both GPT-J and Toolformer on all QA Tasks - despite no access to the WikiSearch tool.

Table 8 shows the performance of the LLaMA-Toolformer model, compared with the original LLaMA base model. It is also compared to a toolformer trained by fine-tuning LLaMA with the original GPT-J Toolformer data. In this we see that both models show improvements using the QA tool on the LAMA tasks, and both show deterioration in using the WikiSearch tool on the QA tasks, although in both tasks, the LLaMA-Toolformer method seems to generally show a greater aptitude with active tool use, than its GPT-J-data counterpart.

### G.4  Discussion

We believe these results indicate some subtleties around how tool-strength, model strength and the synthetic dataset curation interact, and present discussions in the following subsections.

#### G.4.1  Tool Utility

A key factor in explaining the strong performance of the LLaMA-Toolformer of LAMA tasks, but weak performance on QA tasks can be seen by examining the two key tools for each task.

Table 6: Number of final examples, generating on 1% of the CCNet data and scoring with various models and taking the best example per generation location. $k$ represents the maximum number of generation locations per paragraph, and $m$ represents the number of generations per location. Note that fewer GPT-J generations meet the threshold $\tau_f$ when scored by LLaMA instead of GPT-2.

| Generation Model
Scoring Model | GPT-J
GPT-2 | GPT-J
LLaMA | LLaMA
LLaMA |
|---|---|---|---|
| $\tau_f = 0.5, m = 5, tool = $ QA | | | |
| $k = 5$ | 1166 | 308 | 1691 |
| $k = 2$ | 719 | 214 | 1032 |
| $k = 1$ | 98 | 39 | 586 |
| $\tau_f = 0.5, m = 5, tool = $ Wiki | | | |
| $k = 5$ | 1206 | 426 | 1294 |
| $k = 2$ | 753 | 264 | 837 |
| $k = 1$ | 76 | 76 | 463 |
| $\tau_f = 1, m = 5, tool = $ QA | | | |
| $k = 5$ | 437 | 61 | 592 |
| $k = 2$ | 254 | 48 | 392 |
| $k = 1$ | 41 | 19 | 239 |
| $\tau_f = 1, m = 5, tool = $ Wiki | | | |
| $k = 5$ | 324 | 107 | 361 |
| $k = 2$ | 215 | 67 | 212 |
| $k = 1$ | 23 | 22 | 135 |

The QA tool, used in LAMA tasks, consists of a much larger Atlas model that provides a succinct answer to a question, whereas the WikiSearch tool, used in QA tasks, consists of a 'bag of words'-based BM-25 algorithm that retrieves a longer passage from Wikipedia. The former tool is accurate with highly targeted answer, whereas the latter is much noisier and returns a less focused answer. This makes the QA tool far more useful for the format of our evaluation than the WikiSearch tool.

In fact, qualitative investigation revealed that the WikiSearch tool often hindered performance by distracting the model, by frequently the returning passages that did not contain the relevant answer, or may even have concerned a different topic entirely. This was corroborated by the fact that increasing the triggering of API calls was observed to decrease the final scores for all QA tasks, and the LLaMA base model already surpassed the GPT-J Toolformer using the same tool.

The implication of this finding is that tools may vary in utility to different models, and that a big factor in this is the base model strength. A guiding principle should thus be that tool utility will depend on the existing strength of the underlying model on some task. *Strong models need tools that can address their weaknesses.*

### G.4.2 Generating and Scoring

A key feature of the Toolformer method is the scoring of generated data to select how 'useful' a tool call is in lowering the loss of the base model. Our results in Table 8 demonstrate that finetuning a LLaMA model on data generated and scored by itself (LLaMA-Toolformer), slightly outperforms one finetuned on data generated by GPT-J and scored by GPT-2 (LLaMA + GPT-J data).

We investigated how the scoring distributions changed between these two datasets, by performing an ablation in generating and scoring from the same 1% of the CCNet Data. Our results are shown in Table 6.

We find that GPT-J-generated data was much less likely to score highly under LLaMA, though it scored highly under GPT-2. We also found that the LLaMA model was much more capable at generating data when only one sample was taken from the model at each API call point. Both results were expected given the much stronger capabilities of LLaMA v1 7B compared to GPT-2, and seem to suggest a benefit to scoring with the more capable model. *More capable base models benefit from more capable scoring models.*

| Model | LAMA Tasks | | | | QA Tasks | | Maths Tasks | | |
|---|---|---|---|---|---|---|---|---|---|
| | SQuAD | Google-RE | T-REx | WebQS | NQ | TriviaQA | ASDiv | SVAMP | MAWPS |
| GPT-J | 17.8 | 4.9 | 31.9 | 18.5 | 12.8 | 43.9 | 7.5 | 5.2 | 9.9 |
| Toolformer (disabled) | 22.1 | 6.3 | 34.9 | 18.9 | 12.6 | 46.7 | 14.8 | 6.3 | 15.0 |
| Toolformer | **33.8** | **11.5** | **53.5** | 26.3 | 17.7 | 48.8 | 40.4 | 29.4 | 44.0 |
| LLaMA 7B | 27.2 | 1.1 | 40.3 | 21.1 (**36.2**) | 16.9 (**25.4**) | 26.7 (**59.2**) | **41.0** | **34.9** | **48.2** |

Table 7: Results demonstrating the broad ability of the LLaMA v1 7B base model, compared to GPT-J and GPT-J-based Toolformer. Results using the modified prompt from Appendix C.5 are presented in brackets. The LLaMA v1 7B already outperforms the Toolformer in all tasks QA and Maths tasks tool-free, provided the right prompt.

| Model | LAMA Tasks | | | | QA Tasks | |
|---|---|---|---|---|---|---|
| | SQuAD | Google-RE | T-REx | WebQS | NQ | TriviaQA |
| LLaMA v1 7B | 27.2 | 1.1 | 40.3 | 36.2 | 25.4 | 59.2 |
| LLaMA-Toolformer (disabled) | 27.7 | 1.1 | 41.8 | **37.0** | **27.6** | **60.2** |
| LLaMA-Toolformer | **30.0** | **12.3** | **55.5** | 36.6 | 26.9 | 57.8 |
| LLaMA + GPT-J data (disabled) | 27.2 | 1.1 | 40.3 | **37.1** | 27.1 | **59.7** |
| LLaMA + GPT-J data | **28.6** | **11.1** | **54.2** | 35.7 | 26.8 | 55.7 |

Table 8: Results of the Toolformer method applied to the LLaMA base model. "LLaMA-Toolformer" represents a Toolformer trained on LLaMA generated and scored data, "LLaMA + GPT-J data" represents LLaMA fine-tuned on the data used to train the original GPT-J Toolformer. Both toolformers benefit from use of the strong QA tool, and neither benefit from the use of the weak WikiSearchTool tool. Only LLaMA + GPT-J data benefits from the Calculator, since it was trained on an order of magnitude more data - the LLaMA-Toolformer failed to call the tool.

## G.5 Failed Investigations with Calculator

Finally we also investigated experiments with the Calculator tool, rerunning generation with approximately half the original CCNet data. We ran experiments with only this data on account of the extensive compute requirements needed to generate calculator data, which requires many 'repeat attempts' to generate a valid call. Although we generated and scored from half the data, we were only able to generate 401 examples at a $\tau_f$ threshold of 0.5, an order of magnitude less than GPT-J, and from this limited data we were not able to elicit any tool use from the model. We hypothesize that this was down to the fact that appropriate locations for Calculator calls are relatively sparse in CCNet, and the LLaMA models are already highly competent in simple mathematics - outperforming both GPT-J and Toolformer on our benchmark tasks. In future we recommend generating examples from either more maths-based training datasets or human annotations.