# OpenReview forum: "Toolformer: Language Models Can Teach Themselves to Use Tools"
_NeurIPS.cc/2023/Conference — NeurIPS 2023 oral_

### Official Review · Reviewer_Vx6Q · 2023-06-25

**Soundness:** 4 excellent
**Presentation:** 4 excellent
**Contribution:** 3 good
**Rating:** 6
**Confidence:** 5

**Summary:**

The paper explores an interesting area to extend large language models (LLMs) with external tools. The authors show that LLMs can teach themselves to better utilize tools. They tested GPT-j on several tools (calculator, QA system, search engine, translator, and calendar). The experimental results well support the claim, and the model even surpasses GPT-3 despite owning far fewer parameters.

**Strengths:**

+ The paper is well-written and easy to follow.

+ The idea is novel and the supporting experimental results are extensive.

+ The authors study a very interesting topic which I believe will be impactful in the LLM era.

**Weaknesses:**

The experiments are only conducted on GPT-j (a non-instruction tuned model), which I believe is not enough, considering the existence of more powerful open-source LLMs such as LLaMA and Vicuna.

I actually tested that Vicuna, ChatGPT, and GPT-4 already have excellent capabilities in utilizing the tools mentioned in the paper (almost perfect). These models can skillfully manipulate tools based on very simple prompting, achieving far better performance than the reported number in this paper (I'm not sure why this paper only includes GPT-3 as the baseline, which apparently performs poorer than most of the current LLMs). Hence I doubt whether the proposed method could still benefit well-tuned SOTA LLMs.

**Questions:**

NA

---

> ### Author Rebuttal · Authors · 2023-08-09
>
> > The experiments are only conducted on GPT-j (a non-instruction tuned model), which I believe is not enough, considering the existence of more powerful open-source LLMs such as LLaMA and Vicuna.
>
> We completely agree with the reviewer, but at the time during which this work was conducted, neither of the models were available. We are actively working on the experiments with the LLaMA family of models and intend to include those results.
>
> > I actually tested that Vicuna, ChatGPT, and GPT-4 already have excellent capabilities in utilizing the tools mentioned in the paper (almost perfect). These models can skillfully manipulate tools based on very simple prompting, achieving far better performance than the reported number in this paper (I'm not sure why this paper only includes GPT-3 as the baseline, which apparently performs poorer than most of the current LLMs). Hence I doubt whether the proposed method could still benefit well-tuned SOTA LLMs.
>
> It is challenging to respond fully to this question without details about the specific experiment that the reviewer conducted, but it is plausible that models like GPT-4 are capable of utilizing tools when the specific tool description is included in the prompt (and given that the details of this model are not released, it is not implausible that ChatGPT and GPT-4 have already seen tool usage in the pre-training or alignment stage). However, the very act of mentioning the tool potentially hints to the model that it should use a tool, thereby inadvertently simplifying the problem of tool usage. Our approach aims to enable Toolformer to automatically know when to leverage these tools and how. We conjecture that simple prompting is not sufficient when more, possibly redundant, tools become available: verifying this conjecture is part of our future research plans.

---

### Official Review · Reviewer_dirG · 2023-07-05

**Soundness:** 4 excellent
**Presentation:** 3 good
**Contribution:** 4 excellent
**Rating:** 8
**Confidence:** 4

**Summary:**

This paper proposes an approach to augment language models with the ability to call "tools" during decoding, such as a calculator, retrieval system, or machine translation system. This requires only a few human-written examples, and then uses the LM to generate a larger fine-tuning datasets constructed from raw text. When fine-tuned on this dataset, and augmented with the ability to execute external tools, performance of the LM is improved for a range of downstream tasks, across various model scales.

**Strengths:**

* The paper proposes a relatively elegant way to integrate tools with language models in a way that requires only a limited amount of human-written examples of API calls per tool. The proposed method to synthetically construct the fine-tuning dataset appears to work well in practice.
* By showcasing a variety of tools and their impact across a collection of tasks, this paper showcases the potential impact of integrating such tools and their ability to address some common limitations of LMs. The paper seems likely to influence future work.

**Weaknesses:**

* I did not find any significant weaknesses in the proposed approach, execution of the experiments, or technical descriptions in the paper.
* My only gripe is in the wording of the title claim that LMs can "teach themselves to use tools". I can see what the authors mean, as a LM is used to generate the fine-tuning data, but I don't find this to be a helpful description of the method and I think the paper would read better without this bit of hype. Additionally, for new tools, the approach still requires a prompt, a handful of examples, and heuristics for selecting relevant subsets of a corpus. Anyways, this gripe shouldn't be blocking for publication, and I don't expect the authors to change their selected title.



**Questions:**

* The use of a threshold based on the likelihood assigned by the LM with and without the tool use is clever, but I also wonder whether this could be misleading in some cases. For instance, the LM may have been trained (?) on some of the CCNet data, so this may lead to an overly optimistic likelihood without tool usage relative to the optimal tool usage at test time, especially for, .e.g., temporally-sensitive facts. Were any such limitations related to the filtering process observed in practice?

**Limitations:**

Yes

---

> ### Author Rebuttal · Authors · 2023-08-09
>
> > The use of a threshold based on the likelihood assigned by the LM with and without the tool use is clever, but I also wonder whether this could be misleading in some cases. For instance, the LM may have been trained (?) on some of the CCNet data, so this may lead to an overly optimistic likelihood without tool usage relative to the optimal tool usage at test time, especially for, e.g., temporally-sensitive facts. Were any such limitations related to the filtering process observed in practice?
>
> We agree that when the LM has been trained on some data, this may lead to an overly optimistic likelihood without tool usage. We actually don't see this necessarily as a limitation: for some data/knowledge already in the model's weights, it is positive to not call a tool but rely on its weights. Conversely, when the model "knows it doesn't know", which is the case with well calibrated probabilities, we want it to call the tool. This is quite analogous to humans that would rely on, for example, a calculator, for complex calculus. If the training data covers a sufficiently long time span, temporally-sensitive questions will tend to have more uncertain answers and would fall into this category. We agree that more quantitative analysis would be interesting to measure such behavior.

---

> > ### Comment · Reviewer_dirG · 2023-08-16
> >
> > Thank for your response! I have read the response and the other reviews and confirm my original rating.

---

### Official Review · Reviewer_6zEw · 2023-07-06

**Soundness:** 4 excellent
**Presentation:** 3 good
**Contribution:** 3 good
**Rating:** 7
**Confidence:** 4

**Summary:**

This paper proposes an innovative method for enabling Language Models (LMs) to utilize tools. The authors prompt the LM to generate API calls based on human demonstrations, which are then executed in tools. Any non-contributing API calls are filtered out. A dataset is then augmented with these API calls, and used to fine-tune the LM. The Toolformer surpasses larger models in many tasks, offering a significant contribution to the field.

**Strengths:**

This paper outlines a remarkably simple yet effective strategy for curating a dataset that empowers LMs to utilize tools. The method is well-explained and detailed, boasting a universal applicability across multiple datasets and tools. The authors have carried out extensive, well-designed experiments that showcase the performance boost facilitated by their method. The comparison experiment involving Toolformer, a disabled Toolformer, and GPT-J+CC is particularly commendable, as it eliminates the potential of additional fine-tuning data contributing to performance improvement.

This research addresses a practical and intriguing topic that is likely to attract considerable interest from both the research and industrial communities. The potential to integrate more sophisticated tools and utilize larger LMs holds promise for advancing LM capabilities.


**Weaknesses:**

The proposed method has some limitations. First, there's a dependency on fine-tuning when adapting the LM to new tools, which could impede broad usage and necessitate additional work.

Secondly, the use of square brackets for the "<API>" token, without any special escaping mechanism, might present issues when square brackets form part of the original text.

Lastly, the MLQA experiment raises a few questions. The performance of OPT(66B) and GPT-3(175B) suffers due to their inability to provide answers in English, suggesting a potentially inappropriate evaluation setting. Toolformer also underperforms GPT-J in certain languages, seemingly due to the impact of fine-tuning on CCNet. However, it's unclear why Toolformer lags behind GPT-J+CC in German and Arabic. The MLQA experiment fails to convincingly support the paper's main claims.

Additionally, the paper lacks an analysis of cases where the LM fails to use tools effectively during inference. For instance, the reasons behind the LM's failure when using a calculator during inference are not investigated. Is it due to the inability to generate the <API> token or the candidate? Or does it fail to provide the correct answers even when the API call is successful? An examination of these failed cases and the issues causing them would be enlightening.


**Questions:**

To summarize the inquiries raised in the Weaknesses section:

Does the use of square brackets as API tokens interfere with the standard usage of square brackets in the text?

Why does Toolformer underperform in comparison to GPT-J+CC in German and Arabic in the MLQA experiment?

Under what circumstances does Toolformer fail to utilize tools effectively?

---

> ### Author Rebuttal · Authors · 2023-08-09
>
> > The proposed method has some limitations. First, there's a dependency on fine-tuning when adapting the LM to new tools, which could impede broad usage and necessitate additional work.
>
> At the time this work was conducted there was no evidence that effective tool use could be achieved purely through in-context learning. Indeed, in the absence of reproducible descriptions of how models such as ChatGPT and GPT-4, which do exhibit tool use capabilities, were trained, one has to entertain the possibility that they too were fine-tuned towards tool use in some way. Our work introduces a simple, effective, and reproducible way to endow a model that had no tool training with the capability to use tools, something that we believe is relevant to anyone wishing to train their own model.
>
> > Secondly, the use of square brackets for the "<API>" token, without any special escaping mechanism, might present issues when square brackets form part of the original text. Does the use of square brackets as API tokens interfere with the standard usage of square brackets in the text?
>
> This is an understandable concern, however, an API call requires not only the square bracket but also that the specific tool name follows the square bracket (e.g., <API> Calendar() </API>). Consequently, it would be highly unlikely for parts of the original text to be confused with a true API call, but use cases where usage of the square bracket is common would have to consider this issue, which we have added to the Limitations section.
>
> > Why does Toolformer underperform in comparison to GPT-J+CC in German and Arabic in the MLQA experiment?
>
> Our hypothesis is that the GPT-J+CC model already has some knowledge of other languages so often the model doesn’t need to translate the question into English in order to answer it correctly. In fact, performing the API call can sometimes confuse the model due to the special characters, so the model’s answers may be worse in such cases. However, it is not clear why GPT-J+CC performs better for some languages but not others.
>
> From our analysis, the model calls the correct API (MT in this case) with appropriate arguments (either the entire or part of the question) most of the time. The relative poor performance on the MLQA benchmark comes from the inability of answering these questions, even after they’ve been translated into English. However, this is orthogonal to the main goal of the paper which is learning when and how to call a certain API, which we believe our model does to a reasonable extent.
>
> > Under what circumstances does Toolformer fail to utilize tools effectively?
>
> Please see the general response for more details.

---

> > ### Comment · Reviewer_6zEw · 2023-08-17
> >
> > Thank you for your detailed reply. I have no other concerns at this moment.

---

### Official Review · Reviewer_jLhs · 2023-07-07

**Soundness:** 3 good
**Presentation:** 3 good
**Contribution:** 3 good
**Rating:** 7
**Confidence:** 4

**Summary:**

This paper proposes a method to finetune pretrained autoregressive language models such that they learn when and how to use external tools to achieve good performance in downstream tasks. Following an in-context learning scheme, humans provide a few examples of inserting API calls at appropriate location in the natural language sentence, which is ised by the trained language model to automatically generate an API-augmented variation of a natural language corpus. These API augmented sentences are filtered via a threshold over a criterion measuring whether adding the API calls and their result as a prefix improves the perplexity of the natural language sentence under the LM. This filtered API augmented dataset is used for further finetuning of the pretrained LM so that it learns to insert API calls at appropriate places. This approach called toolformer is compared against using the LM alone, finetuning the LM on the natural language corpus, using toolformer but aritificially suppressing its ability to call an API, and larger general purpose language models. 5 different APIs are considered in this setting. The model is evaluated on several downstream tasks for which access to the APIs might be beneficial.

**Strengths:**

-- The paper is very well motivated. This capability of querying external API while generating text is a natural solution to the pathologies like hallucination that the language models exhibit today. This approach is a step toward endowing a language model with such capabilities reliably.

-- The experimental setup is well designed and the choices of APIs, baselines, and downstream tasks to evaluate on lead to informative analysis.

-- This approach outperforms baselines convincingly on the downstream tasks while not drastically affecting the language modeling capabilities as measured by perplexity on the held-out set.

**Weaknesses:**

-- From the writeup, this approach doesn't seem to generate multiple API calls in a sentence and also doesn't perform well with nested API calls. More discussion on this would be useful.

-- While it is discussed in the limitation section and Table 2, a more thorough analysis of sample efficiency of this approach would be helpful. How many sentences are enough to learn tool-use functionality? Is it possible to collect enough high-quality API augmented sentences easily?

-- Ablation study: performance as a function of filtering threshold that controls the quality of the API-augmented sentences would give more insight into the learnability of tool-use and sensitivity to the "correctness" of the API-augmented dataset.

-- A thorough error analysis of failure modes would improve the understanding of the limitations of the proposed approach more clearly.



**Questions:**

See above

**Limitations:**

See above

---

> ### Author Rebuttal · Authors · 2023-08-09
>
> > From the writeup, this approach doesn't seem to generate multiple API calls in a sentence and also doesn't perform well with nested API calls. More discussion on this would be useful.
>
> This is currently touched upon in the Limitations section: “API calls for each tool are generated independently; as a consequence, there are no examples of chained tool use in the finetuning dataset.” We will make clearer that “chained tool use” encompasses multiple API calls.
>
> > While it is discussed in the limitation section and Table 2, a more thorough analysis of sample efficiency of this approach would be helpful. How many sentences are enough to learn tool-use functionality? Is it possible to collect enough high-quality API augmented sentences easily?
> Ablation study: performance as a function of filtering threshold that controls the quality of the API-augmented sentences would give more insight into the learnability of tool-use and sensitivity to the "correctness" of the API-augmented dataset.
>
> While we agree with the reviewer that it would be valuable to quantify how many examples are necessary for tool-use, this undoubtedly varies according to the specific tool. For some tools like a search engine, this may not need that many samples, but the same is not necessarily true for more complex APIs like scheduling a meeting.  Furthermore, finding good opportunities to generate complex examples in a corpus like CCNet poses yet another challenge. The proposed ablation experiments would also require significant compute (at least ten model training ablations for the five tools) and the results may not generalize.
>
> > A thorough error analysis of failure modes would improve the understanding of the limitations of the proposed approach more clearly.
>
> While we did not conduct an extensive and quantitative error analysis, we do have evidence that suggests failure modes are prevalent with some combinations of task and tool. For example, for the DateSet dataset, the Calendar tool is necessary for every example, but we observe that the Toolformer tends to call it less frequently than needed (92.9% of the time). Note however that our evaluation datasets are skewed towards requiring immediate tool use, and that a quantitative error analysis on those would not necessarily shed light on failure patterns in less biased conditions.  Additionally, the task of attributing each incorrect answer across all evaluations to a specific failure mode (failing to call the tool, calling the wrong tool, calling the tool incorrectly, receiving a useless or wrong result from the tool, or failing to come to the wrong conclusion using the tool results) is a major challenge that requires expert human annotation and ideally a broader set of tools.  We leave a general study of failure modes across data sources for future work, and instead include in the camera-ready version a discussion of the different types of failure modes and relevant statistics that we are able to report. Please see the general response for more details.

---

> > ### Comment · Reviewer_jLhs · 2023-08-16
> > **Thanks for the response**
> >
> > I am keeping my initial score.

---

### Author Rebuttal · Authors · 2023-08-09

We thank the reviewers for their time and efforts to review, discuss and improve the paper. We have written responses to each reviewer in turn.

Two of the reviewers asked for more details on the types of errors we have seen in the evaluation. While a detailed and quantitative classification of each failure for each task and tool would require expert human annotation, we do notice broad trends. Namely, we observed that failure can arise from the following:

Failing to call the tool:
- Our evaluation and finetuning datasets are distributionally different since the former are all ultra-short form QA style tasks while the latter consists of CCNet paragraphs, with only a few tool calls per paragraph.  This difference in distribution likely leads to under-use of the tool in our desired setting.  We correct for this by triggering a tool-use when a start token is in the top 10 tokens for the Toolformer, but this is clearly an example of ‘failing to call the tool’. We note that tools are not called for every question, especially where the answer is strongly in-weights.

Calling the wrong tool:
- We find Toolformer very often calls an appropriate tool and can often judge the context correctly. In the maths section, we do observe occasional calls to Question Answering and WikiSearch likely because examples using these tools are much more frequent in the fine-tuning dataset than those using the calculator tool.

Calling the tool incorrectly.
- Many of the tools we use cannot be called ‘incorrectly’, taking either no arguments or strings as inputs - all of which are valid as API calls.
In the case of the Calculator tool, at data-augmentation time, we see many incorrect/invalid generations (even with constrained decoding to arithmetic tokens), but since the final dataset contains useful, correct API calls, the fine-tuned model often generates valid calculations. However, these calls are often very simple (often two number +,-, /, *), and have low complexity.

Receiving the wrong result from the tool or failing to come to the right conclusion
- In some cases we see that a tool response is either useless or incorrect - most often this can be seen with the WikiSearch tool which uses a naive BM25 information retrieval algorithm on Wikipedia.
- The model’s response to this varies and sometimes ignores the result of the tool, while at other times incorporates it.  More investigation is needed to understand _when_ or _why_ a tool is ignored.
Anecdotally, we observe that the natural continuation requires a highly specific answer, but the tool has not returned it - for instance  “Harry Styles was born in [WikiSearch(Harry styles) -> Harry Styles is an British singer and actor, who has starred in My Policeman] Worcestershire.”

---

### Decision · Program_Chairs · 2023-09-21

**Decision:**

Accept (oral)

**Comment:**

This paper proposes augmenting LLMs with the ability to call "tools" (e.g., as a calculator), which is done by providing an initial (small) set of examples, which is then used to augment more examples by leveraging LLM to generate a larger fine-tuning datasets for "instruction tuning" for tool-use.

The proposed technique (cascade of steps) is straightforward, in an elegant way, which also enables the model to generalize showcased by the capability of using a wide range of tools.

All the reviewers agreed the value this paper adds and acknowledged the potential impact of the paper. Accept as a spotlight.